# An Efficient Bit-Based Approach for Mining Skyline Periodic Itemset Patterns

Yanzhi Li [1,2] and Zhanshan Li [2,3,*]

1  College of Software, Jilin University, Changchun 130012, China; yanzhi21@mails.jlu.edu.cn
2  Key Laboratory of Symbolic Computation and Knowledge Engineering of Ministry of Education, Jilin University, Changchun 130012, China
3  College of Computer Science and Technology, Jilin University, Changchun 130012, China
*  Correspondence: lizs@jlu.edu.cn

**Abstract:** Periodic itemset patterns (PIPs) are widely used in predicting the occurrence of periodic events. However, extensive redundancy arises due to a large number of patterns. Mining skyline periodic itemset patterns (SPIPs) can reduce the number of PIPs and guarantee the accuracy of prediction. The existing SPIP mining algorithm uses FP-Growth to generate frequent patterns (FPs), and then identify SPIPs from FPs. Such separate steps lead to a massive time consumption, so we propose an efficient bit-based approach named BitSPIM to mine SPIPs. The proposed method introduces efficient bitwise representations and makes full use of the data obtained in the previous steps to accelerate the identification of SPIPs. A novel cutting mechanism is applied to eliminate unnecessary steps. A series of comparative experiments were conducted on various datasets with different attributes to verify the efficiency of BitSPIM. The experiment results demonstrate that our algorithm significantly outperforms the latest SPIP mining approach.

**Keywords:** data mining; periodic pattern; skyline periodic itemset pattern; bitwise operation

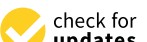



## 1. Introduction

Data mining plays a significant role in data analysis and knowledge extraction [1]; it has become an efficient tool for pattern discovery due to its applicability in a variety of circumstances such as association rule mining (ARM) [2], clustering analysis [3], and classification [4]. Mining frequent patterns (FPs) [2] are fundamental in ARM. The connection-based algorithm, called Apriori [5], is a classical breadth-first iterative algorithm for mining FPs. Many algorithms have been developed to accelerate the mining of FP. Han et al. proposed a depth-first algorithm called FP-Growth [6,7], based on FP-tree. It uses a prefix tree structure without generating candidates and only scans the dataset twice. BitTableFI [8], as proposed by Dong et al., employs an efficient bit structure to compress the dataset.

After the proposal of ARM, many new types of patterns have emerged, including high-utility patterns [9], periodic itemset patterns (PIPs) [10], subgraph patterns [11], and sequential patterns [12], etc. Among them, PIPs are one of the most well-studied types of patterns. For instance, the opportunity for online or offline retailers to recommend suitable products to their customers is very critical, because the right recommendation may satisfy the customers, while a completely wrong one may be a turnoff to the customers. Customers may buy a new product when the old one reaches its expected life or is consumed, therefore, it is safe to assume that there is a relationship between the lifespan or consumption cycle of a product and its purchase number and cycle. By tapping into the purchase frequency and period of a product in customers' shopping records, retailers cannot only improve the shopping experience of customers but also allow themselves to better understand the buying habits of customers, raise the recommendation hit rates, promote similar products, increase user stickiness, and so on. Accordingly, when the criteria of frequency and period

are considered together, the retailers can make advisable marketing strategies. Therefore, it is very necessary to utilize the periodic itemset patterns in the shopping records in the decision-making department of retailers.

PIPs can be used to predict the occurrence of periodic events [13], deal with the seasonality information of products [14], and serve in recommendation systems [15]. PIPs consider both the frequency and periodicity of an itemset and are regarded as an expanded derivative of FPs. There are various periodicity measures for PIPs [16], which lead to different definitions, including the maximum period [17], variance of periods [18], and so on. In 2021, Chen et al. adopted a measure based on the coefficient of variation to define PIPs [19]. In their work, an itemset is a PIP if its coefficient of variation is less than or equal to the threshold of coefficient of variation, indicating that the fluctuation of the period of the itemset is below the average level. They proposed a probability model for predicting periodic patterns. The frequency and periodicity influence the prediction accuracy of the probability model. For an itemset, a higher frequency indicates a wider range of sample sizes of the periods, and a lower coefficient of variation means less fluctuation. The model is limited due to the redundancies originating from predicting items that are contained in different PIPs multiple times. The redundancies are proportional to the number of PIPs.

In 2023, Chen et al. proposed a special sort of PIP, named the Skyline Periodic Itemset Pattern (SPIP) [20], aimed at making accurate pattern predictions. In SPIPs, PIPs with either higher frequency or lower coefficient of variation, or both, are preferred. They provided the definition of SPIP and proposed an effective algorithm named SPIM for mining SPIPs. Patterns that are not dominated by any other patterns in two dimensions constitute the skyline of a 2-dimensional dataset [21]. A PIP is an SPIP if there are no other PIPs with both higher frequency and lower coefficient of variation. By mining SPIPs, we can significantly reduce the number of patterns while ensuring the accuracy of predictions. The aim of mining SPIPs is to avoid a vast number of PIPs and relieve users from an overload of patterns.

SPIM is divided into two steps. The first step is to mine all FPs in advance using FP-Growth, and then identify SPIPs from the FPs obtained in the first step. Using FP-Growth to mine FPs makes SPIM consist of two very independent steps. Additionally, the occurrence sets of an itemset are generated in the second step, even if the itemset has already been identified as an FP. Confined by these two complicated stages, SPIM consumes massive computational resources. The running time of SPIM is longer than that of FP-Growth, as FP-Growth essentially serves as part of SPIM. In terms of memory usage, constructing FP-trees in FP-Growth consumes significant memory resources. These disadvantages of SPIM motivate the development of a more efficient SPIP mining approach.

Instead of using separate steps, we found that the identification of SPIP can proceed as soon as an itemset is recognized as an FP. Additionally, efficient bitwise representations can accelerate set operations. We present a novel approach called Bitwise Skyline Periodic Itemset Pattern Mining (BitSPIM) for mining SPIPs. This method utilizes bitwise representations in an Apriori-like algorithm named BitTableFI [8] to deal with FPs while incorporating a novel cutting mechanism. Once an itemset is recognized as an FP, the bitset for its occurrence set is directly used to derive its period list and coefficient of variation, which are then used to determine whether the itemset is an SPIP. Simulated experiments were conducted on ten transaction datasets with divergent characteristics to compare the performance of BitSPIM and SPIM. The experimental results demonstrate the effectiveness of the proposed method in terms of running time and memory usage. We believe that BitSPIM could be an influential alternative in mining SPIPs.

## 2. Related Works

In this section, we review related works and techniques concerning mining SPIPs. SPIP is a special type of PIP. In the field of PIP mining, different periodicity measures can lead to various types of PIPs. Maximum period [17] can be used as the periodicity measure for PIPs, and such PIPs are mined by periodic frequent pattern growth, which utilizes a

tree structure. Fournier-Viger et al. provided various kinds of periodic measures. Three measures named minimum periodicity, maximum periodicity, and average periodicity are proposed in [22], and an algorithm named Periodic Frequent Pattern Miner mines PIPs with the aid of the monotonicity of these three types of periodicity. Additionally, they introduced the definitions of periodic standard deviation and sequence periodic ratio [23] to mine PIPs common to multiple sequences. A regularity measure for PIPs is defined using the variance of periods [18]. Based on the standard deviation, the coefficient of variation is adopted to measure PIPs in the works of Chen et al. [19]. They then inherited the coefficient of variation measure to define SPIP in [20].

Mining FPs is a fundamental procedure in mining SPIPs. Depth-first search and breadth-first search are two main methods for mining FPs, known as candidate generation and pattern growth, respectively [24]. Depth-first algorithms search for FPs in a bottom-up manner. Starting from itemsets containing a single item, larger FPs with more items are recursively generated by appending items according to the total order. Han et al. proposed a depth-first algorithm called FP-Growth [6,7], based on the FP-tree, to compress database transactions. This method consumes a significant amount of running time in creating multiple subtrees. Additionally, the performance of the algorithm is affected by the storage consumption from recording a substantial number of FP-tree nodes.

As for breadth-first search, Apriori [5] proposed by Agrawal et al., is a classical breadth-first FP mining algorithm. It is a fundamental iterative algorithm that uses a layer-by-layer search to find FPs, employing an iterative search pattern and a test-and-generate approach. Based on the Apriori algorithm, several algorithms have been developed to compress the database, allowing for the quick generation of candidate itemsets and the calculation of their support. T-Apriori [25] uses an overlap strategy when counting support to ensure high efficiency. BitTableFI [8], proposed by Dong et al., employs an efficient bit structure to compress the database.

Apart from approaches like BitTableFI for mining FPs, bitwise representations and operations are exploited in various works in mining metadata. Index-BitTableFI [26] is an improved version of BitTableFI, which utilizes heuristic information provided by an index array. SPAM [27], aimed at mining sequential patterns, employs a bitmap representation of the database. In IndiBits [28], proposed by Breve et al., the binary representation of data similarities is used, and bitwise operations are employed to update the Binary Attribute Satisfiability (BAS) Distance Matrix. For mining frequent closed itemsets, algorithms for efficiently calculating the intersection between two dynamic bit vectors [29] are proposed. CloFS-DBV [30] also utilizes dynamic bit vectors to mine frequent closed itemsets. The computation of support is based on dynamic bit vectors when generating new patterns. These bit vectors can also be used in mining web access patterns [31]. Trang et al. proposed two algorithms named MWAPC and EMWAPC, which are based on the prefix-web access pattern tree (PreWAP) structure for mining web access patterns with a super-pattern constraint. In DPMmine [32], vector column intersection bitwise operations are used to aid the algorithm in mining colossal pattern sequences.

## 3. Background and Preliminaries

Let $I = \{i_1, i_2, \ldots, i_m\}$ denote a set of finite items, $|I|$ is the number of items in $I$. The items are discrete real numbers or symbols. As shown in Figure 1, there are mapping relations that map these discrete numbers and symbols into a group of continuous items. In our paper, we assume that there exist mapping relations that map the real numbers or symbols into a series of continuous integers starting from 1. The relevant definitions of mining SPIPs are presented as follows:

**Definition 1.** *A transaction $T_k$ is a set of items in $I$, i.e., $T_k \subseteq I$. $T_k$ holds a unique index k called the transaction identifier.*

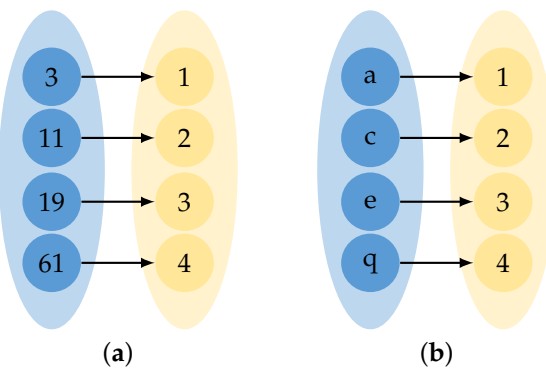

**Figure 1.** The diagram of the mapping relations. (**a**) Map the discontinuous numbers to continuous items. (**b**) Map the symbols to continuous items.

A transaction dataset $DB = \{T_1, T_2, \ldots, T_n\}$ comprises $n$ transactions. $|DB|$ is the number of transactions in $DB$. Table 1 shows an example transaction dataset $DB_1$ containing five transactions denoted by $T_1$ to $T_5$, where $I = \{1, 2, 3, 4, 5\}$, $|DB_1| = 5$. Example 1 shows the relationship between $T_k$ and $I$, where $1 \leq k \leq 5$. Transactions represent a shopping list of products from the retailer that are purchased by a customer; $I$ can be used to represent the whole set of products on the shopping list. The transaction dataset can be extracted from the database of the retailer, which is served as the shopping record in a time interval.

**Table 1.** Example transaction dataset $DB_1$.

| Transaction | Items |
|:---:|:---:|
| $T_1$ | 1, 2, 3, 5 |
| $T_2$ | 1, 3, 4, 5 |
| $T_3$ | 1, 2, 3, 4, 5 |
| $T_4$ | 3, 4, 5 |
| $T_5$ | 2, 3, 4 |

**Example 1.** *For the set of items $T_1 = \{1, 2, 3, 5\}$ in Table 1, since $T_1 \subseteq I$, $T_1$ is a transaction. For another set of items $\{1, 2, 6, 8\}$, which is not a subset of $I$, it is not a transaction.*

**Definition 2.** *An itemset, $X$, is a non-empty set, and $X \subseteq I$. An itemset, $X$, containing $n$ items is called an $n$-itemset. $n$ is the size of the itemset. Specifically, $\{i\}$ is a 1-itemset that contains a single item $i$.*

**Example 2.** *$X_1 = \{3\}$ and $X_2 = \{2, 3\}$ are two itemsets with sizes of 1 and 2. Thus, the two itemsets are also called a 1-itemset and a 2-itemset, respectively.*

**Definition 3.** *The occurrence set $O_X$ for an itemset $X$ is a set of transaction identifiers, $O_X = \{k \mid X \subseteq T_k, T_k \in DB\}$.*

**Example 3.** *In Table 1, $T_1$, $T_3$, and $T_5$ incorporate $X = \{2, 3\}$, so $O_X = \{1, 3, 5\}$.*

**Definition 4.** *The frequency $Freq_X$ for an itemset $X$ is the ratio of the size of $O_X$ to the number of transactions in the dataset, $Freq_X = |O_X|/|DB|$. Given a frequency threshold $\theta$, an itemset $X$ is a frequent pattern if $Freq_X \geq \theta$.*

**Example 4.** *For $DB_1$ in Table 1 with a frequency threshold $\theta = 0.7$, for $X_1 = \{2, 3\}$ and $X_2 = \{3, 4\}$, $O_{X_1} = \{1, 3, 5\}$ and $O_{X_2} = \{2, 3, 4, 5\}$. Thus, $Freq_{X_1} = 0.6$ and $Freq_{X_2} = 0.8$. Since $Freq_{X_2} = 0.8 > 0.6$, $X_2$ is a frequent pattern. Similarly, $X_1$ is not a frequent pattern since $Freq_{X_1} = 0.6 < 0.7$.*

**Definition 5.** *The period list $Per_X$ for an itemset X is the set of periods of X: $Per_X = \{w_{p+1} - w_p \mid \forall p \in \{1, \ldots, |O_X| - 1\}, w_p \in O_X\}$.*

**Definition 6.** *The coefficient of variation $C_X$ of an itemset X is the ratio of the standard deviation of $Per_X$ to the mean of $Per_X$: $C_X = std(Per_X) / mean(Per_X)$. $std(*)$ and $mean(*)$ represent the standard deviation and mean, respectively.*

**Example 5.** *$X_1 = \{1, 2\}$ and $X_2 = \{3, 6\}$ in $DB_2$, as shown in Table 2, $O_{X_1} = \{1, 4, 6, 8, 11\}$. Thus by Definition 5, $Per_{X_1} = \{3, 2, 2, 3\}$. The standard deviation and the mean of $Per_{X_1}$ are 0.5 and 2.5, respectively, $C_{X_1} = std(Per_{X_1}) / mean(Per_{X_1}) = 0.2$. Similarly, $O_{X_2} = \{3, 4, 5, 12\}$ and $Per_{X_2} = \{1, 1, 7\}$, $C_{X_2} = 0.943$.*

**Table 2.** Example transaction dataset $DB_2$.

| Transaction | Items | Transaction | Items |
|---|---|---|---|
| $T_1$ | 1, 2, 5, 6, 8 | $T_7$ | 2, 3, 4 |
| $T_2$ | 1, 4, 5, 6 | $T_8$ | 1, 2, 3, 7 |
| $T_3$ | 1, 3, 6, 7 | $T_9$ | 3, 5, 7, 8 |
| $T_4$ | 1, 2, 3, 6 | $T_{10}$ | 2, 3, 4 |
| $T_5$ | 3, 4, 6, 8 | $T_{11}$ | 1, 2, 7, 8 |
| $T_6$ | 1, 2, 6, 7 | $T_{12}$ | 3, 4, 6, 7, 8 |

The coefficient of variation is a suitable metric for measuring the periodicity of patterns [19]. It reflects the fluctuation in the appearance of patterns in the transaction dataset. Patterns with a lower coefficient of variation exhibit better periodicity, while a higher coefficient of variation indicates irregularity in occurrence. We follow the approach of Chen et al. in introducing the coefficient of variation as a measure of periodicity [19].

**Definition 7.** *For a transaction dataset, a frequency threshold $\theta$, and a coefficient of variation threshold $\delta$, an itemset X is a periodic itemset pattern if X is a frequent pattern and $C_X \leq \delta$. The set of PIPs is denoted by PIP:*

$$PIP = \{X \mid X \in FP, C_X \leq \delta\}.$$

**Example 6.** *$X_1 = \{1, 2\}$ and $X_2 = \{3, 6\}$ in $DB_2$, as shown in Table 2, with a frequency threshold $\theta = 0.2$ and a coefficient of variation threshold $\delta = 0.5$. Both $X_1$ and $X_2$ are FPs, as their frequencies are beyond 0.2. As $C_{X_1} = 0.2 < 0.5$, by Definition 7, $X_1$ is a PIP. Similarly, $X_2$ is not a PIP since $C_{X_2} = 0.943 > 0.5$.*

**Definition 8.** *For two itemsets (X and Y) in a transaction dataset, X is dominated by Y if $Freq_X < Freq_Y$ and $C_X \geq C_Y$, or $Freq_X \leq Freq_Y$ and $C_X > C_Y$. 'X is dominated by Y' is equivalent to 'Y dominates X'.*

**Example 7.** *For $X_1 = \{1, 2\}$, $X_2 = \{3, 6\}$, and $X_3 = \{3\}$ in $DB_2$, as shown in Table 2, the frequency and the coefficient of variation of $X_1$, $X_2$, and $X_3$ are listed in Table 3. By Definition 8, neither $X_2$ nor $X_3$ dominate $X_1$, as $C_{X_1} < C_{X_2}$ and $C_{X_1} < C_{X_3}$. Neither $X_1$ nor $X_2$ dominate $X_3$ as $Freq_{X_3} > Freq_{X_1}$ and $Freq_{X_3} > Freq_{X_2}$. As $Freq_{X_2} < Freq_{X_1}$ and $C_{X_2} > C_{X_1}$, $X_2$ is dominated by $X_1$. Similarly, it is dominated by $X_3$.*

**Table 3.** The frequency and the coefficient of variation of $X_1$, $X_2$, and $X_3$ in $DB_2$.

| Pattern | Frequency | Coefficient of Variation |
|---|---|---|
| $X_1$ | 0.416 | 0.2 |
| $X_2$ | 0.333 | 0.943 |
| $X_3$ | 0.667 | 0.351 |

**Definition 9.** *For a transaction dataset, DB, a frequency threshold, θ, and a coefficient of variation threshold, δ, an itemset, X, is an SPIP if X is a periodic itemset pattern and X is not dominated by other itemsets in DB. The set of SPIPs is denoted by SPIP:*

$$SPIP = \{X \mid X \in PIP, \nexists\, Y \text{ s.t. } Y \text{ dominates } X\}.$$

By Definitions 8 and 9, the aim of mining SPIPs is to explore the patterns that are more frequent or have better periodicity or both.

## 4. BitSPIM: The Proposed Method

*4.1. The Preliminaries of Bitwise Representation*

In our approach, bitsets and efficient bitwise representations are introduced to deal with set operations.

**Definition 10.** *The bitset for a set, X, is denoted by $\mathcal{BS}_X$. $\mathcal{BS}_X[i]$ is the ith bit of $\mathcal{BS}_X$. If an item $i \in X$, then $\mathcal{BS}_X[i]$ is assigned as **1**. Otherwise, it is assigned as **0**:*

$$\mathcal{BS}_X[i] = \begin{cases} \mathbf{1} & i \in X \\ \mathbf{0} & i \notin X \end{cases}$$

*The **Set** operation and **Clear** operation are used to assign **1** and **0** to the bits in the bitset, respectively.*

$|X|$ and $|\mathcal{BS}_X|$ are the sizes of $X$ and $\mathcal{BS}_X$, respectively. $|\mathcal{BS}_X|$ equals the number of bits assigned, as **1** in $\mathcal{BS}_X$. Obviously, $|\mathcal{BS}_X| = |X|$. By Definition 10, a mapping relation between the set and its bitwise representation is established. This relation enables the efficient use of bitwise operations when handling sets. For example, the intersection operation and union operation between sets are equivalent to performing "&" and "|" on their bitsets, respectively.

**Definition 11.** *The value of a bitset $\mathcal{BS}_X$ denoted by $V_{\mathcal{BS}_X}$ is the binary number of $\mathcal{BS}_X$.*

As shown in Example 8, the bitsets can be regarded as binary numbers; thus, the value of bitsets can directly be compared.

**Example 8.** *For $X_1$ = {2, 3, 5} and $X_2$ = {3, 4, 5} in $DB_1$, $\mathcal{BS}_{X_1}$ and $\mathcal{BS}_{X_2}$ are **01101** and **00111**, respectively. $V_{\mathcal{BS}_{X_1}} > V_{\mathcal{BS}_{X_2}}$ as **01101 > 00111**.*

The transactions are also sets of items. If an item $i$ is in a transaction $T_k$, $\mathcal{BS}_{T_k}[i]$ is assigned as **1**. Hereby, the bitset for a transaction is obtained. For a dataset $DB$, the bitwise representation of $DB$ is derived by obtaining the bitsets for all transactions. The bitwise representation of $DB_1$ is shown in Table 4.

**Table 4.** Bitwise representation of $DB_1$.

| Transaction | Items | $\mathcal{BS}_{T_k}$ |
|:-----------:|:-----:|:--------------------:|
| $T_1$ | 1, 2, 3, 5 | *11101* |
| $T_2$ | 1, 3, 4, 5 | *10111* |
| $T_3$ | 1, 2, 3, 4, 5 | *11111* |
| $T_4$ | 3, 4, 5 | *00111* |
| $T_5$ | 2, 3, 4 | *01110* |

**Definition 12.** *The head of an itemset X denoted by $head_X$ is the minimal item in X, it corresponds to the first **1** bit in $\mathcal{BS}_X$. Accordingly, the tail of an itemset, X, denoted by $tail_X$, is the maximal item in X, and it corresponds to the last **1** bit in $\mathcal{BS}_X$.*

**Example 9.** *As shown in Figure 2, for the 3-itemsets $X_1$ = {2, 3, 4} and $X_2$ = {2, 3, 5} in $DB_1$, $head_{X_1}$ = $head_{X_2}$ = 2, $tail_{X_1}$ = 4 and $tail_{X_2}$ = 5.*

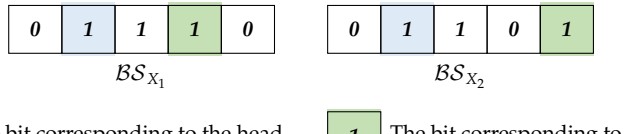

**Figure 2.** Diagram of Example 9. The bits corresponding to the head and the tail of $X_1$ and $X_2$ are colored, respectively.

**Definition 13.** *Given a transaction dataset, DB, and its bitwise representation, I is the set of items in DB; for an item $i \in I$, the column $Col_i$ for i is the bitset for the occurrence set $O_{\{i\}}$, where $Col_i = \mathcal{BS}_{O_{\{i\}}}$.*

By Definitions 3 and 10, with $\mathcal{BS}_{O_X}$, the frequency of $X$ can be calculated as $|\mathcal{BS}_{O_X}|$ = $|O_X|$. For an itemset $X$, Algorithm 1 shows the procedures to obtain the bitset for $O_X$. Initially, $\mathcal{BS}_{O_X}$ equals $Col_{head_X}$ (line 4). Then, $\mathcal{BS}_{O_X}$ is obtained by performing bitwise "**&**" operations on the columns for other items in $X$ (lines 5 to 9). The worst time complexity of Algorithm 1 is $\mathcal{O}(|I|^2/64)$, where $|I|$ is the number of items in the dataset. Example 10 provides an illustration of acquiring $\mathcal{BS}_{O_X}$ for an itemset $X$ in Table 1.

---

**Algorithm 1** GetOccur

---

1: **Input**: $\mathcal{BS}_X$: **bitset**
2: **Output**: $\mathcal{BS}_{O_X}$: **bitset**
3: $head_X \leftarrow$ the head of $X$
4: $\mathcal{BS}_{O_X} \leftarrow Col_{head_X}$
5: **for each** $i \in \mathcal{BS}_X$ **do**
6:     **if** $i > head_X$ **then**
7:         $\mathcal{BS}_{O_X} \leftarrow \mathcal{BS}_{O_X}$ **&** $Col_i$
8:     **end if**
9: **end for**
10: **return** $\mathcal{BS}_{O_X}$

---

**Example 10.** *By Table 4, for an itemset $X$ = {2, 4} in Table 1, $Col_2$ = **10101**, $Col_4$ = **01111**. As shown in Figure 3, by performing "&" on $Col_2$ and $Col_4$, $\mathcal{BS}_{O_X}$ is **00101**.*

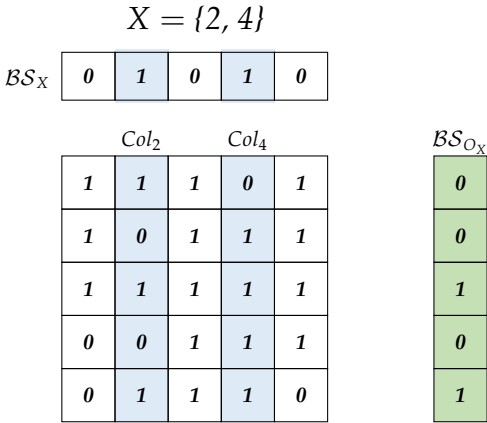

Bitwise representation of $DB_1$

**Figure 3.** Diagram of Example 10. Given the bitwise representation of $DB_1$, $X$ = {2, 4} and $\mathcal{BS}_X$ = **01010**. $Col_2$, $Col_4$, and $\mathcal{BS}_{O_X}$ are colored. $\mathcal{BS}_{O_X}$ = $Col_2$ & $Col_4$ = **00101**.

**Definition 14.** *For an itemset, X, the prefix for X is denoted by $\mathcal{P}_X$. It is a bitset equal to $\mathcal{BS}_X$ while the last **1** bit is **Cleared**.*

For two *k*-itemsets, *X* and *Y*, if *X* and *Y* have the same prefix, they have $k - 1$ items in common and can be merged into a new $(k + 1)$-itemset *Z*. By Definition 12, the two *k*-itemsets, *X* and *Y*, and the new $(k + 1)$-itemset, *Z*, have an identical head, and the tail of *Z* is the larger one between $tail_X$ and $tail_Y$. Example 11 provides an illustration.

**Example 11.** *As shown in Figure 4, for the 3-itemsets $X_1$ = {2, 3, 4} and $X_2$ = {2, 3, 5} in $DB_1$, since $\mathcal{P}_{X_1} = \mathcal{P}_{X_2}$ = **01100**, by merging $X_1$ and $X_2$, a new 4-itemset $X_3$ is generated and $\mathcal{BS}_{X_3}$ = **01111**, $head_{X_3} = head_{X_1} = head_{X_2} = 2$. As $tail_{X_2} > tail_{X_1}$, $tail_{X_3} = tail_{X_2} = 5$.*

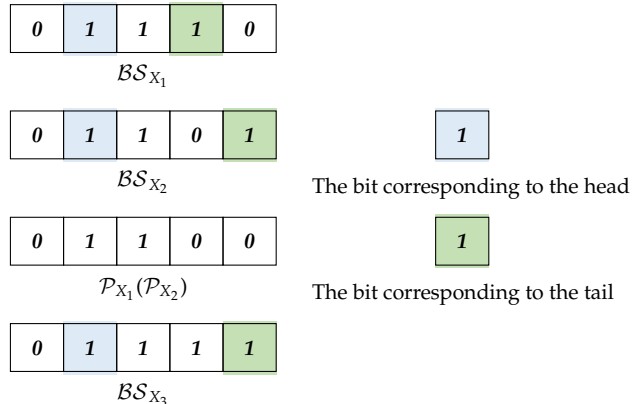

**Figure 4.** Diagram of Example 11. The bitset for $X_1$, $X_2$, and $X_3$, as well as the prefix for $X_1$ and $X_2$ are depicted. The bits corresponding to the head and the tail of the itemset are colored, respectively.

In this paper, we specify that only itemsets with the same prefix can be merged.

*4.2. Our Theories and Data Structure*

Based on the aforementioned preliminary definitions and concepts, we introduce the critical knowledge and basic data structure to induce our proposed method. In BitSPIM, SPIPs are identified iteratively. We mark the iteration that generates the SPIPs of size *k* as *k*th iteration.

**Definition 15.** *Given dataset DB, X and Y are two itemsets in DB, $|X| = |Y|$. If $V_{\mathcal{BS}_X} > V_{\mathcal{BS}_Y}$, then $\mathcal{BS}_X \succ \mathcal{BS}_Y$.*

$\succ$ reflects the relative position of the bitsets. If $\mathcal{BS}_X \succ \mathcal{BS}_Y$, $\mathcal{BS}_Y$ is after $\mathcal{BS}_X$. Obviously, the transitivity of $\succ$ between bitsets is satisfied. For three bitsets, $\mathcal{BS}_X$, $\mathcal{BS}_Y$, and $\mathcal{BS}_Z$, if $\mathcal{BS}_X \succ \mathcal{BS}_Y$ and $\mathcal{BS}_Y \succ \mathcal{BS}_Z$, then $\mathcal{BS}_X \succ \mathcal{BS}_Z$.

**Corollary 1.** *For two bitsets, $\mathcal{BS}_X$ and $\mathcal{BS}_Y$, if $\mathcal{BS}_X \succ \mathcal{BS}_Y$, then $V_{\mathcal{P}_X} \geq V_{\mathcal{P}_Y}$.*

**Proof.** We denote $V_{tail_X}$ as the value of the binary number for the bitset with the only **1** at the tail of *X*, $V_{\mathcal{P}_X} = V_{\mathcal{BS}_X} - V_{tail_X}$, $V_{\mathcal{P}_X} > V_{tail_X}$ and accordingly, $V_{\mathcal{P}_Y} = V_{\mathcal{BS}_Y} - V_{tail_Y}$, $V_{\mathcal{P}_Y} > V_{tail_Y}$. If $\mathcal{BS}_X \succ \mathcal{BS}_Y$, then $|X| = |Y|$ and $V_{\mathcal{BS}_X} > V_{\mathcal{BS}_Y}$. If $V_{tail_X} < V_{tail_Y}$, then $V_{\mathcal{P}_X} > V_{\mathcal{P}_Y}$. If $V_{tail_X} \geq V_{tail_Y}$, assume $V_{\mathcal{P}_X} < V_{\mathcal{P}_Y}$, as $V_{\mathcal{P}_X} > V_{tail_X}$ and $V_{\mathcal{P}_Y} > V_{tail_Y}$, $V_{\mathcal{P}_X} + V_{tail_X} < V_{\mathcal{P}_Y} + V_{tail_Y}$, in other words, $V_{\mathcal{BS}_X} < V_{\mathcal{BS}_Y}$, which is contradictory with $\mathcal{BS}_X \succ \mathcal{BS}_Y$. The assumption is invalid and Corollary 1 is proved. □

**Definition 16.** *The ItemsetList $\mathcal{L}$ is an ordered list; its containing elements are unique bitsets with identical sizes. The $\succ$ relation holds between any two of the bitsets in $\mathcal{L}$.*

PIPs and SPIPs are contained in the sets named $\mathcal{S}_{pip}$ and $\mathcal{S}_{slp}$, respectively. The notations and functions of the ItemsetLists and sets in BitSPIM are shown in Table 5:

**Table 5.** The notations and functions of different ItemsetLists and sets in BitSPIM.

| Notation | Function |
|---|---|
| $\mathcal{S}_{slp}$ | containing SPIPs |
| $\mathcal{S}_{pip}$ | containing PIPs |
| $\mathcal{L}_{cur}$ | representing the bitsets to $k$th iteration |
| $\mathcal{L}_{next}$ | transferring the bitsets to $(k+1)$th iteration |

**Theorem 1.** *Suppose $\mathcal{BS}_X$ and $\mathcal{BS}_Y$ are two bitsets in $\mathcal{L}$ and $\mathcal{BS}_X \succ \mathcal{BS}_Y$. If $\mathcal{P}_X \neq \mathcal{P}_Y$, then there exists no $\mathcal{BS}_Z$, such that $\mathcal{P}_X = \mathcal{P}_Z$ and $\mathcal{BS}_Y \succ \mathcal{BS}_Z$.*

**Proof.** Since $\mathcal{BS}_X \succ \mathcal{BS}_Y$, $V_{\mathcal{P}_X} \geq V_{\mathcal{P}_Y}$ by Corollary 1. As $\mathcal{P}_X \neq \mathcal{P}_Y$, there is

$$V_{\mathcal{P}_X} > V_{\mathcal{P}_Y}. \tag{1}$$

Suppose there exists $\mathcal{BS}_Z$, such that $\mathcal{P}_X = \mathcal{P}_Z$ and $\mathcal{BS}_Y \succ \mathcal{BS}_Z$, then $V_{\mathcal{P}_Y} \geq V_{\mathcal{P}_Z}$ and $V_{\mathcal{P}_X} = V_{\mathcal{P}_Z}$, there is

$$V_{\mathcal{P}_Y} \geq V_{\mathcal{P}_X}. \tag{2}$$

Obviously, (1) and (2) contradict each other. Consequently, Theorem 1 is proved. □

Theorem 1 is the basic efficient cutting mechanism. An illustration of Theorem 1 is provided in Example 12.

**Example 12.** *By Table 1, for five itemsets, $X_1 = \{1, 2, 3\}$, $X_2 = \{1, 2, 5\}$, $X_3 = \{1, 3, 4\}$, $X_4 = \{1, 3, 5\}$, and $X_5 = \{2, 4, 5\}$ in $DB_1$, their bitsets and prefixes are shown in Figure 5. $\mathcal{BS}_{X_1}$ to $\mathcal{BS}_{X_5}$ are contained in $\mathcal{L}$ and there is $\mathcal{BS}_{X_1} \succ \mathcal{BS}_{X_2} \succ \mathcal{BS}_{X_3} \succ \mathcal{BS}_{X_4} \succ \mathcal{BS}_{X_5}$. According to Theorem 1, since $\mathcal{P}_{X_1} \neq \mathcal{P}_{X_3}$, neither the prefix for $X_4$ nor that of $X_5$ equals $\mathcal{P}_{X_1}$. As depicted in Figure 5, different types of bitsets are colored with different colors, respectively. $\mathcal{BS}_{X_1}$ and the bitsets that have the same prefix with $\mathcal{BS}_{X_1}$ are marked in blue; the first bitset that has a different prefix with $\mathcal{BS}_{X_1}$ is marked in green, and the bitsets that are not processed according to Theorem 1 are marked in gray.*

| 11100 | 11001 | 10110 | 10101 | 01011 |
|---|---|---|---|---|
| $\mathcal{BS}_{X_1}$ | $\mathcal{BS}_{X_2}$ | $\mathcal{BS}_{X_3}$ | $\mathcal{BS}_{X_4}$ | $\mathcal{BS}_{X_5}$ |

**Figure 5.** Diagram of Example 12. The bitsets in $\mathcal{L}$ are presented. Different types of bitsets are colored with different colors.

### 4.3. Mining SPIPs Efficiently

In this section, a detailed illustration of BitSPIM is provided. We demonstrate our proposed method with an example of mining SPIPs in the $DB_2$ dataset, as shown in Table 2, with a frequency threshold $\theta = 0.4$. For simplicity, the coefficient of variation threshold $\delta$ is set to $\infty$, which implies that all FPs are also PIPs.

#### 4.3.1. Identification of SPIPs with Bitset

We follow the key steps of the identification of SPIPs described in [20] while several modifications are adopted. According to Chen et al., the identification of SPIPs does not proceed until all FPs are obtained, and at that moment, the occurrence set of each itemset is discovered.

Rather than acquiring all FPs in advance before the identification of SPIPs, in BitSPIM, once an itemset, $X$, is recognized as an FP, the identification of whether $X$ is an SPIP is

executed immediately. The bitset for $O_X$, denoted by $\mathcal{BS}_{O_X}$, can be directly utilized, which has already been obtained when calculating $Freq_X$. The steps of judging whether an FP is an SPIP are described in Algorithms 2 and 3. The function of Algorithm 2 is to remove all itemsets in $\mathcal{S}_{slp}$ that are dominated by an itemset $X$. Suppose $|S_{slp}|$ is the maximal number of itemsets in $S_{slp}$; the worst time complexity of Algorithm 2 is $\mathcal{O}(|S_{slp}|)$. $Freq_{max}$ and $C_{min}$ record the current maximal frequency and the minimal coefficient of variation of the itemsets in $\mathcal{S}_{slp}$, respectively. The steps of Algorithm 3 are as follows:

---

**Algorithm 2** ClearNonSPIP

---

1: **Input:** $\mathcal{S}_{slp}$: **Set**, $X$: **Itemset**
2: **Output:** $\mathcal{S}_{slp}$: **Set**
3: **for each** $Y \in \mathcal{S}_{slp}$ **do**
4:      **if** $Y$ is dominated by $X$ **then**
5:          $\mathcal{S}_{slp} \leftarrow \mathcal{S}_{slp} - \{Y\}$
6:      **end if**
7: **end for**
8: **return** $\mathcal{S}_{slp}$

---

**Algorithm 3** CheckSPIP

---

1: **Input:** $\mathcal{BS}_X$: **Bitset**, $\mathcal{BS}_{O_X}$: **Bitset**, $\delta$: **Double**, $\mathcal{S}_{pip}$: **Set**, $\mathcal{S}_{slp}$: **Set**
2: **Output:** $\mathcal{L}_{cur}$: **List**
3: $Per_X \leftarrow$ the period list of $X$
4: $C_X \leftarrow$ the coefficient of variation of $X$
5: **if** $C_X > \delta$ **then**
6:      **return** $\mathcal{S}_{slp}, \mathcal{S}_{pip}$
7: **end if**
8: $\mathcal{S}_{pip} \leftarrow \mathcal{S}_{pip} \cup \{X\}$
9: **if** $Freq_{max} < Freq_X, C_{min} > C_X$ **then**
10:      $Freq_{max} = Freq_X, C_{min} = C_X$
11:      $\mathcal{S}_{slp} \leftarrow \{X\}$
12: **else**
13:      **if** $Freq_{max} < Freq_X$ **then**
14:          $Freq_{max} = Freq_X$
15:      **else if** $C_{min} > C_X$ **then**
16:          $C_{min} = C_X$
17:      **else if** $\exists Y \in S_{slp}$ **s.t.** $Y$ dominates $X$ **then**
18:          **return** $\mathcal{S}_{slp}, \mathcal{S}_{pip}$
19:      **end if**
20:      $\mathcal{S}_{slp} \leftarrow$ **call** Algorithm 2 ($\mathcal{S}_{slp}, X$)
21:      $\mathcal{S}_{slp} \leftarrow \mathcal{S}_{slp} \cup \{X\}$
22: **end if**
23: **return** $\mathcal{S}_{slp}, \mathcal{S}_{pip}$

---

(1) With $\mathcal{BS}_{O_X}$, by Definitions 5 and 6, $Per_X$ and $C_X$ are acquired (lines 3 to 4), respectively.
(2) If $C_X > \delta$, by Definition 7, $X$ is not a PIP, and the algorithm terminates (line 6). Otherwise, $X$ is added to $\mathcal{S}_{pip}$ (line 8).
(3) If $Freq_{max} < Freq_X$ and $C_{min} > C_X$, by Definition 7, $X$ dominates all itemsets in $\mathcal{S}_{slp}$. Therefore, $X$ is the only element in $\mathcal{S}_{slp}$; the value of $Freq_{max}$ and the value of $C_{min}$ are updated with $Freq_X$ and $C_X$, respectively (lines 9 to 11).
(4) If $Freq_{max} < Freq_X$ and $C_{min} \leq C_X$, or, $Freq_{max} \geq Freq_X$ and $C_{min} > C_X$, $X$ may dominate some itemsets in $\mathcal{S}_{slp}$ and none of the itemsets in $\mathcal{S}_{slp}$ can dominate $X$. $\mathcal{S}_{slp}$ contains $X$ and the itemsets that are not dominated by $X$. Specifically, in the former case, the value of $Freq_{max}$ is updated with $Freq_X$, and in the latter case, the value of $C_{min}$ is updated with $C_X$ (lines 13 to 16 and lines 20 to 21).

(5)   If $Freq_{max} \geq Freq_X$ and $C_{min} \leq C_X$, $X$ may be dominated by some itemsets in $\mathcal{S}_{slp}$. If any itemset dominates $X$ (line 17), $X$ is not an SPIP and the identification of $X$ stops (line 18), $X$ is not in $\mathcal{S}_{slp}$. Otherwise, $\mathcal{S}_{slp}$ contains $X$ and the itemsets that are not dominated by $X$ (lines 20 to 21).

In Algorithm 3, as Algorithm 2 is invoked and $Per_X$ is utilized, the worst-case time complexity of Algorithm 3 is $\mathcal{O}(max\{|I|, |S_{slp}|\})$, where $|S_{slp}|$ represents the maximal number of itemsets in $S_{slp}$.

### 4.3.2. First Iteration

The aim of first iteration is to generate bitsets for frequent 1-itemsets and identify SPIPs of size 1 (if any). Algorithm 4 illustrates the process of first iteration. $I$ is the set of items in the transaction dataset. To guarantee the $\succ$ relation between any two bitsets in the ItemsetLists, the items in $I$ are in ascending order. Initially, the values of $Freq_{max}$ and $C_{min}$ are set to 0 and $\infty$, respectively (line 3). $\mathcal{L}_{cur}$, $\mathcal{S}_{pip}$, and $\mathcal{S}_{slp}$ are empty (line 4). For each item $i$ in $I$, all bits in $\mathcal{BS}_{\{i\}}$ are **Cleared** except that the $i$th bit is set to **1** (lines 6 to 7). Then, by Algorithm 1, $\mathcal{BS}_{O_{\{i\}}}$ is formulated on line 8. As $\mathcal{BS}_{\{i\}}$ contains one **1** bit, the process of lines 5 to 9 in Algorithm 1 is omitted. $Freq_{\{i\}}$ is computed by Definition 4 (line 9). If $Freq_{\{i\}}$ is not less than the frequency threshold $\theta$, $\{i\}$ is an FP and $\mathcal{BS}_{\{i\}}$ is added to the end of $\mathcal{L}_{cur}$ (line 11). Algorithm 3 is then invoked to identify whether $\{i\}$ is an SPIP, as discussed in Section 4.3.1.

---

**Algorithm 4** First iteration

---

1:  **Input**: $I$: **Set**, $\theta$: **Double**, $\delta$: **Double**
2:  **Output**: $\mathcal{L}_{cur}$
3:  $Freq_{max} = 0$, $C_{min} = \infty$
4:  $\mathcal{L}_{cur} \leftarrow$ an empty **List**, $\mathcal{S}_{pip} \leftarrow \varnothing$, $\mathcal{S}_{slp} \leftarrow \varnothing$
5:  **for each** $i \in I$ **do**
6:      $\mathcal{BS}_{\{i\}} \leftarrow$ an empty **Bitset**
7:      **Set** $\mathcal{BS}_{\{i\}}[i]$
8:      $\mathcal{BS}_{O_{\{i\}}} \leftarrow$ **call** Algorithm 1($\mathcal{BS}_{\{i\}}$)
9:      $Freq_{\{i\}} \leftarrow |\mathcal{BS}_{O_{\{i\}}}| / |DB|$
10:     **if** $Freq_{\{i\}} \geq \theta$ **then**
11:         Add $\mathcal{BS}_{\{i\}}$ to the end of $\mathcal{L}_{cur}$
12:         **call** Algorithm 3 ($\mathcal{BS}_{\{i\}}$, $\mathcal{BS}_{O_{\{i\}}}$, $\mathcal{S}_{pip}$, $\mathcal{S}_{slp}$, $\delta$)
13:     **end if**
14: **end for**
15: **return** $\mathcal{L}_{cur}$

---

When Algorithm 4 stops, all infrequent 1-itemsets are eradicated and will not be involved in the subsequent iterations. $\mathcal{L}_{cur}$ becomes the input to the second iteration. In Algorithm 4, Algorithms 1 and 3 are invoked for each item $i$ in $I$. Thus, the worst time complexity of Algorithm 4 is $\mathcal{O}(|I| * (max\{|I|, |S_{slp}|\} + |I|^2/64))$.

An illustration of first iteration is provided for mining SPIPs in the $DB_2$ dataset, as shown in Table 2, with a frequency threshold $\theta = 0.4$ and the coefficient of variation threshold $\delta = \infty$. Table 6 shows the frequencies and the coefficients of variation for all eight 1-itemsets in $DB_2$, denoted by $\{1\}$ to $\{8\}$.

On line 5 of Algorithm 4, the items in $I$ are in ascending order, the bitsets for all 1-itemsets, $\{1\}$ to $\{8\}$, are sequentially processed by Algorithm 4. As the threshold of the coefficient of variation is set to $\infty$, the coefficients of variation for all 1-itemsets are not larger than $\infty$. Consequently, lines 5 to 7 of Algorithm 3 are skipped. Initially, for $\mathcal{BS}_{\{1\}}$, as $Freq_{max} = 0$ and $C_{min} = \infty$, $\{1\}$ is added to $S_{slp}$, $Freq_{max} = 0.583$ and $C_{min} = 0.447$. As $Freq_{max} = Freq_{\{2\}}$ and $C_{min} = C_{\{2\}}$, lines 17 to 21 of Algorithm 3 are used to process $\{2\}$; $\{2\}$ can also be added to $S_{slp}$ as $\{1\}$ does not dominate $\{2\}$. $Freq_{max}$ and $C_{min}$ remain invariant.

As $Freq_{max} < Freq_{\{3\}}$ and $C_{min} > C_{\{3\}}$, lines 9 to 11 of Algorithm 3 are used to process $\{3\}$; $\{3\}$ dominates $\{1\}$ and $\{2\}$ and is added to $S_{slp}$ while $\{1\}$ and $\{2\}$ are removed from $S_{slp}$. $Freq_{max} = 0.667$ and $C_{min} = 0.351$. As $Freq_{max} > Freq_{\{4\}}$ and $C_{min} > C_{\{4\}}$, lines 15 to 16 and 20 to 21 of Algorithm 3 are used to process $\{4\}$; $\{3\}$ stays in $S_{slp}$ as it is not dominated by $\{4\}$. After $\{4\}$ is processed, $S_{slp}$ contains $\{3\}$ and $\{4\}$, $Freq_{max} = 0.667$ and $C_{min} = 0.2$. As $Freq_{\{5\}}$ is less than the frequency threshold, $\{5\}$ is not an SPIP as it is not an FP (line 10 of Algorithm 4). For itemsets $\{6\}$ to $\{8\}$, their frequencies are less than $Freq_{max}$. Moreover, they can be dominated by some itemsets in $S_{slp}$ (line 17 of Algorithm 3). At the end of 1st iteration, $\{3\}$ and $\{4\}$ are two SPIPs. According to lines 10 to 11 of Algorithm 4, $\mathcal{L}_{cur}$ contains the bitsets for $\{1\}$ to $\{8\}$ except $\{5\}$, as the frequency of $\{5\}$ is less than $\theta$. $\mathcal{L}_{cur}$ is then used as the input to the second iteration.

**Table 6.** The frequency and the coefficient of variation of eight 1-itemsets in $DB_2$. $Freq_{max}$, $C_{min}$, and $S_{slp}$ denote the maximal frequency, the minimal coefficient of variation, and the set of SPIPs after itemset $\{i\}$ is processed. $I$ is the set of items in $DB_2$, $i \in I$.

| Itemset ($\{i\}$) | Frequency | Coefficient of Variation | $Freq_{max}$ | $C_{min}$ | $S_{slp}$ |
|---|---|---|---|---|---|
| $\{1\}$ | 0.583 | 0.447 | 0.583 | 0.447 | $\{1\}$ |
| $\{2\}$ | 0.583 | 0.447 | 0.583 | 0.447 | $\{1\}, \{2\}$ |
| $\{3\}$ | 0.667 | 0.351 | 0.667 | 0.351 | $\{3\}$ |
| $\{4\}$ | 0.416 | 0.2 | 0.667 | 0.2 | $\{3\}, \{4\}$ |
| $\{5\}$ | 0.25 | 0.75 | 0.667 | 0.2 | $\{3\}, \{4\}$ |
| $\{6\}$ | 0.583 | 1.016 | 0.667 | 0.2 | $\{3\}, \{4\}$ |
| $\{7\}$ | 0.5 | 0.415 | 0.667 | 0.2 | $\{3\}, \{4\}$ |
| $\{8\}$ | 0.416 | 0.472 | 0.667 | 0.2 | $\{3\}, \{4\}$ |

4.3.3. $k$th Iteration ($k > 1$)

As shown in Algorithm 5, in $k$th iteration, SPIPs of size $k$ are obtained, and frequent $(k+1)$-itemsets are generated and used as the input to $(k+1)$th iteration. $k$th iteration activates as $\mathcal{L}_{cur}$ covers the bitsets for all frequent $(k-1)$-itemsets. The procedures of Algorithm 5 are as follows:

(1)　When $\mathcal{L}_{cur}$ is not empty, Algorithm 5 runs iteratively (line 3).
(2)　$\mathcal{L}_{next}$ is set to empty (line 4).
(3)　For each $\mathcal{BS}_X$ in $\mathcal{L}_{cur}$, $\mathcal{P}_X$ is preliminarily constructed (line 6). According to Definition 14, $\mathcal{P}_X$ is equal to $\mathcal{BS}_X$ while the last **1** bit is substituted by **0** (lines 7 to 8).
(4)　To generate new $(k+1)$-itemsets, for each $\mathcal{BS}_Y$ after $\mathcal{BS}_X$ in $\mathcal{L}_{cur}$, if $\mathcal{P}_X$ differentiates from $\mathcal{P}_Y$, all bitsets after $\mathcal{BS}_Y$ have a different prefix compared to that of $\mathcal{BS}_X$, according to Theorem 1; thus, no bitset can be combined with $\mathcal{BS}_X$. Therefore, none of the bitsets after $\mathcal{BS}_Y$ will be further processed while determining which bitsets can be merged with $\mathcal{BS}_X$ (line 11). Otherwise, $X$ and $Y$ can be merged as they share an identical prefix. This approach of limiting the traversal of bitsets avoids extensive, pointless searches on itemsets that are inevitably unable to be merged.
(5)　When $\mathcal{BS}_Y$ processes an identical prefix, the last bit that indicates the tail is the only discrepancy between them. The combination of $\mathcal{BS}_X$ and $\mathcal{BS}_Y$ focuses on the last 1-bit rather than trivially performing a bitwise "|" operation on $\mathcal{BS}_X$ and $\mathcal{BS}_Y$. A new bitset $\mathcal{BS}_N$ is constructed for the $(k+1)$-itemset, which initially equals $\mathcal{BS}_X$ (line 13).
(6)　The $tail_Y$th bit in $\mathcal{BS}_N$ is **set** to **1** (line 14).
(7)　Resembles 1st iteration, $Freq_N$ is calculated by Algorithm 1 and Definition 4 (lines 15 and 16).
(8)　If $Freq_N$ is greater than or equal to the frequency threshold, $\mathcal{BS}_N$ is added to the end of $\mathcal{L}_{next}$ (line 18).
(9)　With $\mathcal{BS}_N$ and $\mathcal{BS}_{O_N}$, Algorithm 3 is invoked to examine whether itemset $N$ is an SPIP (line 19).

(10) While $\mathcal{L}_{next}$ covers the bitsets for all $(k+1)$-itemsets, the bitsets in $\mathcal{L}_{next}$ are transferred to $\mathcal{L}_{cur}$ (line 23). This step declares both the end of $k$th iteration and the beginning of $(k+1)$th iteration.

---

**Algorithm 5** $k$th iteration $(k > 1)$

---

1: **Input:** $\mathcal{L}_{cur}$: **List**, $\theta$: **Double**, $\delta$: **Double**
2: **Output:** $\mathcal{L}_{cur}$: **List** $\mathcal{S}_{slp}$: **Set**, $\mathcal{S}_{pip}$: **Set**
3: **while** $\mathcal{L}_{cur}$ is not an empty **List do**
4:     $\mathcal{L}_{next} \leftarrow$ an Empty **List**
5:     **for each** $\mathcal{BS}_X \in \mathcal{L}_{cur}$ **do**
6:         $\mathcal{P}_X \leftarrow \mathcal{BS}_X$
7:         $tail_X \leftarrow$ the tail of $X$
8:         ***Clear*** $\mathcal{P}_X[tail_X]$
9:         **for each** $\mathcal{BS}_Y \in \mathcal{L}_{cur}$ and $\mathcal{BS}_X \succ \mathcal{BS}_Y$ **do**
10:             **if** $\mathcal{P}_X \neq \mathcal{P}_Y$ **then**
11:                 **break**
12:             **end if**
13:             $\mathcal{BS}_N \leftarrow$ **clone** $\mathcal{BS}_X$
14:             ***Set*** $\mathcal{BS}_N[tail_Y]$
15:             $\mathcal{BS}_{O_N} \leftarrow$ **call** Algorithm 1($\mathcal{BS}_N$)
16:             $Freq_N \leftarrow |\mathcal{BS}_{O_N}| / |DB|$
17:             **if** $Freq_N \geq \theta$ **then**
18:                 Add $\mathcal{BS}_N$ to the end of $\mathcal{L}_{next}$
19:                 **call** Algorithm 3 ($\mathcal{BS}_N$, $\mathcal{BS}_{O_N}$, $\mathcal{S}_{pip}$, $\mathcal{S}_{slp}$, $\delta$)
20:             **end if**
21:         **end for**
22:     **end for**
23:     $\mathcal{L}_{cur} \leftarrow \mathcal{L}_{next}$
24: **end while**
25: **return** $\mathcal{L}_{cur}$, $\mathcal{S}_{slp}$, $\mathcal{S}_{pip}$

---

When $\mathcal{L}_{cur}$ is an empty list, no frequent $(k+1)$-itemset is generated in $k$th iteration, $(k+1)$th iteration will not proceed, and the algorithm terminates; all SPIPs are identified.

Suppose $|\mathcal{L}_{cur}|$ is the maximal number of bitsets in $\mathcal{L}_{cur}$, the worst time complexity of an arbitrary $k$th iteration is $\mathcal{O}(|\mathcal{L}_{cur}|^2 * (\ max\{|I|, |S_{slp}|\} + |I|^2/64))$.

We provide an illustration of 2nd iteration for mining SPIPs in the $DB_2$ dataset, as shown in Table 2 with a frequency threshold $\theta = 0.4$ and the coefficient of variation threshold $\delta = \infty$. $\mathcal{L}_{cur}$ contains the bitset for $\{1\}$, $\{2\}$, $\{3\}$, $\{4\}$, $\{6\}$, $\{7\}$, and $\{8\}$. Algorithm 3 only checks if $X_1 = \{1, 2\}$ and $X_2 = \{1, 6\}$ are SPIPs, as among all the 2-itemsets, only $X_1$ and $X_2$ are FPs with a frequency beyond $\theta$. For simplicity, Table 7 merely gives the frequency and the coefficient of variation of $X_1$ and $X_2$ in Table 2.

At the beginning of 2nd iteration, $Freq_{max} = 0.667$ and $C_{min} = 0.2$. As $Freq_{X_1} < Freq_{max}$ and $C_{X_1} = C_{min}$, lines 17 to 21 of Algorithm 3 are used to process $X_1$. Neither $\{3\}$ nor $\{4\}$ dominates $X_1$ and $X_1$ cannot dominate $\{3\}$ or $\{4\}$; thus. $\{3\}$, $\{4\}$, and $X_1 = \{1, 2\}$ are SPIPs. $Freq_{max}$ and $C_{min}$ remain invariant. Similarly, for $X_2 = \{1, 6\}$, as $Freq_{X_2} \leq Freq_{\{4\}}$ and $C_{X_2} > C_{\{4\}}$, $X_2$ is dominated by $\{4\}$; thus, it is not an SPIP. At the end of 2nd iteration, $\mathcal{S}_{slp}$ contains three SPIPs: $\{3\}$, $\{4\}$, and $\{1, 2\}$. $\mathcal{L}_{cur}$ contains two bitsets for $X_1$ and $X_2$, which are used as the inputs of 3rd iteration.

In 3rd iteration, only a bitset for a 3-itemset $X_3 = \{1, 2, 6\}$ can be merged. As $X_3$ is not an FP, $\mathcal{L}_{cur}$ is an empty list at the end of 3rd iteration (line 4 and line 23 of Algorithm 5). 4th iteration starts with an empty $\mathcal{L}_{cur}$, the algorithm terminates as 4th iteration stops (line 3 of Algorithm 5), and the final SPIPs in Table 2 with $\theta = 40\%$ and $\delta = \infty$ are $\{3\}$, $\{4\}$ and $\{1, 2\}$.

**Table 7.** The frequency and the coefficient of variation of $X_1$ and $X_2$ in $DB_2$. $Freq_{max}$, $C_{min}$, and $S_{slp}$ show the maximal frequency, the minimal coefficient of variation, and the set of SPIPs after itemset $X_i$ is processed.

| Itemset ($X_i$) | Frequency | Coefficient of Variation | $Freq_{max}$ | $C_{min}$ | $S_{slp}$ |
|---|---|---|---|---|---|
| $X_1 = \{1, 2\}$ | 0.416 | 0.2 | 0.667 | 0.2 | $\{3\}, \{4\}, \{1, 2\}$ |
| $X_2 = \{1, 6\}$ | 0.416 | 0.346 | 0.667 | 0.2 | $\{3\}, \{4\}, \{1, 2\}$ |

## 5. Empirical Evaluation

We conducted a series of experiments to compare the performances of BitSPIM and SPIM on a Windows 10 PC equipped with an AMD Ryzen 3950X processor, with 64 GB of memory. The CPU clock speed is locked to 3.5 GHz to avoid the adverse effects of CPU overclocking. The characteristics of the datasets involved in our experiments are presented in Table 8, including four synthetic datasets and six real datasets. All datasets in the experiments are downloaded from the website SPMF (http://www.philippe-fournier-viger.com/spmf, accessed on 1 September 2023).

As SPIM [20] is the state-of-the-art and the only algorithm focusing on mining SPIPs, we primarily compare the running time and memory usage between our approach and SPIM. All datasets used in SPIM are included in our experiments. Additionally, as FP-Growth is a fundamental component of SPIM, the running time of FP-Growth is also considered to further explore the effectiveness of BitSPIM. For simplicity, all $\delta$ in our experiment are set to $\infty$, which implies that all frequent patterns are also periodic itemset patterns. In the first experiment, the numbers of PIPs and SPIPs identified by both algorithms were recorded. The second experiment focuses on the running time of BitSPIM, SPIM, and FP-Growth. Finally, we compare the performance in terms of memory usage between BitSPIM and SPIM.

**Table 8.** The characteristics of the empirical datasets.

| Dataset | # Trans | # Items | AveLen | Density |
|---|---|---|---|---|
| T10I4D100K | 100,000 | 870 | 10 | 1.15% |
| T20I6D100K | 99,922 | 893 | 19.9 | 2.23% |
| T25I10D10K | 9976 | 929 | 24.77 | 2.67% |
| C20D10K | 10,000 | 192 | 20 | 10.42% |
| Chainstore | 1,112,949 | 46,086 | 7.23 | 0.02% |
| Foodmart | 4141 | 1559 | 4.42 | 0.28% |
| OnlineRetail | 541,909 | 2603 | 4.37 | 0.17% |
| Kosarak | 990,002 | 41,270 | 8.1 | 0.02% |
| BMS-WebView-1 | 59,602 | 497 | 2.51 | 0.51% |
| BMS-WebView-2 | 77,512 | 3340 | 4.62 | 0.14% |

"#" represents "the number of", "Trans" represents "Transactions", "AveLen" represents "Average Length".

### 5.1. Number of Patterns

To verify that the SPIPs obtained by the proposed method are complete and correct, we counted the number of PIPs and SPIPs obtained by BitSPIM and SPIM. The results show that, on all datasets involved in the experiment, the PIP and SPIP numbers mined by BitSPIM are always consistent with those obtained by SPIM for various values $\theta$, verifying the correctness of the proposed method.

### 5.2. Running Time

The running time of BitSPIM is compared with that of SPIM and FP-Growth. In SPIM, FPs are identified in advance using FP-Growth before the recognition of SPIPs; thus, the running time of FP-Growth can be recorded. Figure 6 demonstrates the running times of BitSPIM, SPIM, and FP-Growth on different datasets with various frequency thresholds $\theta$ when $\delta = \infty$. In each subfigure representing the running time on different datasets, the

range of $\theta$ includes the approximate frequency threshold value, where BitSPIM and SPIM have the same running times. The horizontal axis indicates the value of $\theta$, and the vertical axis represents the running time. The red curve, blue curve, and gray curve indicate the running times of BitSPIM, SPIM, and FP-Growth, respectively. The circles on the red curve, the triangles on the blue curve, and the squares on the gray curve represent the running times of our method and that of SPIM and FP-Growth on the specific $\theta$, respectively. The intersection points of the red and blue curves mean the running times of BitSPIM and SPIM are identical. This is projected on the horizontal axis by a dotted line parallel to the vertical axis. The horizontal coordinate of the intersection point indicates the frequency threshold at which the two algorithms have the same running time.

As shown in Figure 6, except at the smaller thresholds, BitSPIM outpaces SPIM in terms of running time across most of the threshold ranges. The curve of the running time for BitSPIM is steeper than that of SPIM. Observing the gradient of the running time curve, as $\theta$ increases, once $\theta$ goes beyond the horizontal coordinate of the intersection point of BitSPIM's curve and SPIM's curve, the running time of BitSPIM is consistently less than that of SPIM. For example, as shown in Figure 6g, the horizontal coordinate of the intersection point is 0.215% on the OnlineRetail dataset. It can be concluded that BitSPIM runs faster than SPIM at 99.785% of the threshold range.

The improvement achieved by BitSPIM over SPIM with respect to running time is significant on datasets T20I6D100K, Chainstore, OnlineRetail, and Kosarak. For example, on the T20I6D100K dataset, when the frequency threshold is 0.3%, BitSPIM is approximately 2 times faster than SPIM. For frequency thresholds beyond 0.6%, SPIM takes at least 4 times longer than BitSPIM. On datasets T25I10D10K, C20D10K, Foodmart, and BMS-Webview-1, although the improvement is not as pronounced, BitSPIM still shows an advantage over SPIM on the majority of frequency thresholds. Since BitSPIM utilizes the basic idea of Apriori, it is acknowledged that BitSPIM can be outpaced by SPIM at small frequency thresholds. In fact, the experimental results support the conclusion of [33] that no algorithm is an absolute and clear winner, able to outperform all others across all datasets and the entire range of thresholds. Overall, BitSPIM is observed to require less running time compared with SPIM for the majority of frequency thresholds

Mining FPs is fundamental to the identification of SPIPs. SPIM identifies SPIPs from all FPs mined by FP-Growth, and as a result, SPIM naturally takes longer to run than FP-Growth. However, BitSPIM does not adopt separate steps to mine FPs and can demonstrate better performance compared with FP-Growth. On datasets like T20I6D100K, Chainstore, OnlineRetail, and Kosarak, BitSPIM runs faster than FP-Growth for the majority of frequency thresholds. Although on datasets such as T10I4D100K, Foodmart, and BMS-Webview-2, BitSPIM does not show much superiority over FP-Growth, it can still be observed that there is an intersection between the red curve and gray curve, representing the running times of BitSPIM and FP-Growth, respectively. This indicates that BitSPIM can outperform FP-Growth at some frequency thresholds. The comparison between the running times of BitSPIM and FP-Growth further demonstrates the superior performance of our approach over SPIM.

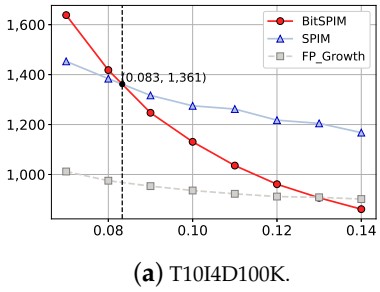

(**a**) T10I4D100K.

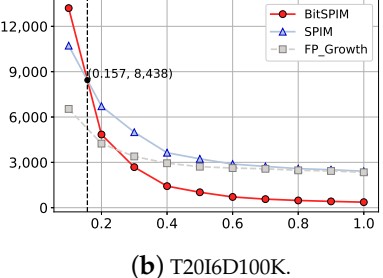

(**b**) T20I6D100K.

**Figure 6.** *Cont.*

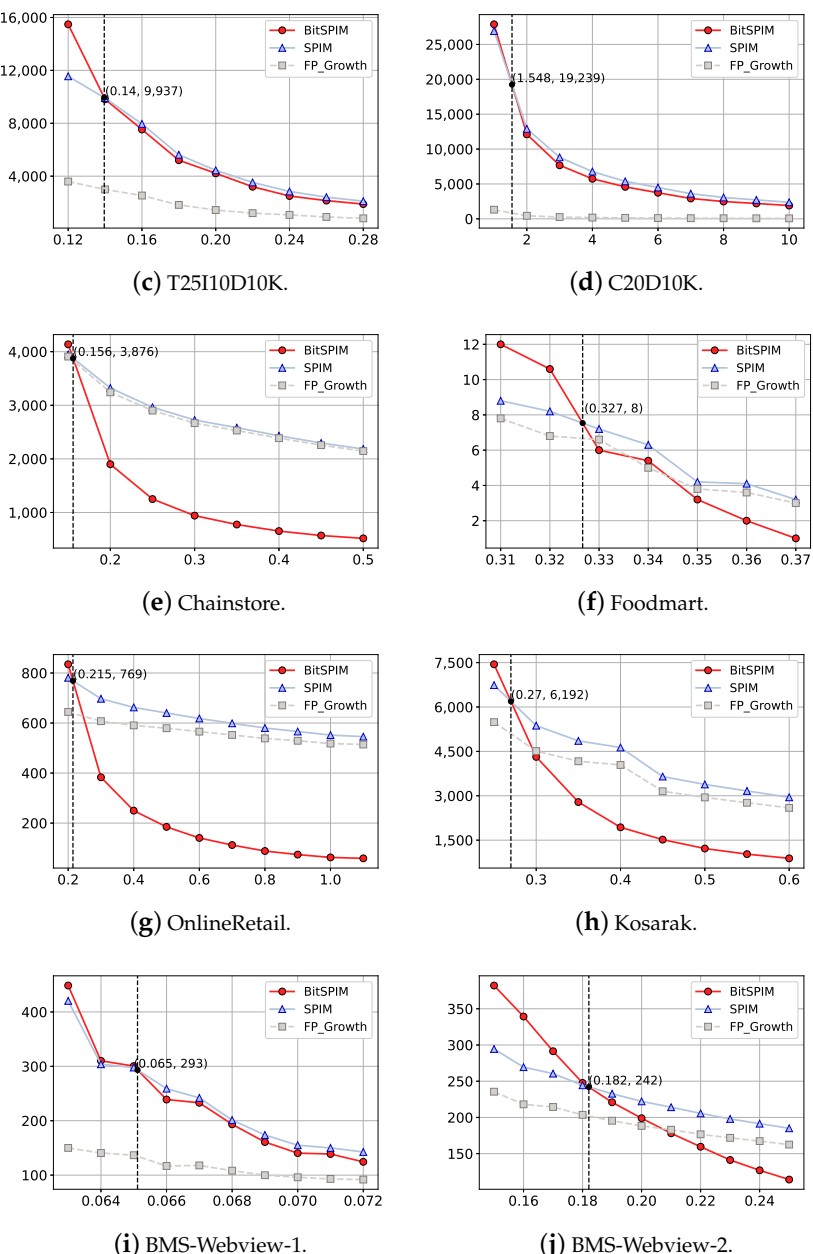

**Figure 6.** Running time (ms) with different frequency thresholds $\theta$ (%) on empirical datasets. The horizontal axis and the vertical axis in each subfigure represent the value of $\theta$ and the running time, respectively. The intersection points of the red and blue curves in each subfigure are projected on the horizontal axis by a dotted line parallel to the vertical axis.

*5.3. Memory Usage*

The results comparing the average memory usage of BitSPIM and SPIM with different frequency thresholds $\theta$ on empirical datasets are presented in Table 9. The better results are highlighted in bold. The coefficient of variance threshold $\delta$ is set to $\infty$ and the same range of frequency thresholds as in the running time experiment are adopted. As shown in Table 9, except on datasets with a large number of transactions and items, such as Chainstore and Kosarak, BitSPIM outperforms SPIM in terms of average memory usage.

**Table 9.** Average memory usage (MB) of BitSPIM and SPIM on empirical datasets. The better result in each row is marked in bold.

| Dataset | SPIM | BitSPIM |
|---|---|---|
| T10I4D100K | 3021.2 | **247.1** |
| T20I6D100K | 3496.2 | **381.8** |
| T25I10D10K | 5237.8 | **2255.2** |
| C20D10K | 4360.1 | **2072.8** |
| Chainstore | **5371.6** | 6558.1 |
| Foodmart | 4308.7 | **1670.0** |
| OnlineRetail | 5286.1 | **2936.9** |
| Kosarak | **4622.1** | 5667.4 |
| BMS-WebView-1 | 5112.3 | **1697.8** |
| BMS-WebView-2 | 4781.4 | **1805.1** |

*5.4. Discussion*

From the results, the proposed method shows better performance, as it consumes less time compared with SPIM for the vast majority of frequency threshold values across different datasets. Regarding memory usage, BitSPIM generally consumes less memory than SPIM, except on datasets with an extensive number of transactions and items.

The advantages of the proposed method can be summarized as follows: (1) The bitset representation of the transaction dataset is more compact than the original dataset. (2) Bitwise operations are involved in mining SPIPs by mapping ordinary sets to bitsets. The generation of new itemsets and the calculation of their frequency can be realized by performing efficient bitwise operations. (3) A novel cutting technique avoids many unnecessary operations. When certain conditions are met, the loop stops without exploring the entire search space. (4) The off-the-shelf occurrence set of the itemset can be utilized directly when identifying whether an FP is an SPIP. (5) Space for constructing FP-trees is saved as FP-Growth is not used in identifying FPs.

However, due to the inherent drawbacks originating from Apriori, BitSPIM repeatedly scans the dataset to generate new bitsets and calculate the frequency of the itemsets. This leads to higher time consumption at smaller thresholds. On datasets with numerous transactions and items, a large number of bitsets need to be stored and operated in BitSPIM; thus, in such cases, it is outperformed by SPIM in terms of memory usage.

**6. Conclusions**

In this paper, we propose a more efficient approach for mining SPIPs, called BitSPIM, compared with the SPIM algorithm. Apart from utilizing a novel bitwise representation that is capable of mining SPIPs, BitSPIM adopts a cutting mechanism to reduce the search space. We evaluate the performance of our approach in comparison with the latest algorithm for mining SPIPs on a variety of real and synthetic datasets. The results demonstrate that BitSPIM is faster and consumes less memory than SPIM in most cases. We believe that our approach is a significant alternative in mining SPIPs and can be applied to diverse fields within ARM.

**Author Contributions:** Y.L. implemented the experiment and wrote the first draft of the paper, Z.L. provided funding for the paper and revised it. All authors have read and agreed to the published version of the manuscript.

**Funding:** This work is supported by the National Natural Science Foundation of China under grant no. 62276060, Development and Reform Committee Foundation of Jilin province of China under grant no. 2019C053-9.

**Data Availability Statement:** The datasets are available at the following links: http://www.philippe-fournier-viger.com/spmf (accessed on 1 September 2023).

**Conflicts of Interest:** The authors declare no conflict of interest.

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
