# Peer review of "An Efficient Bit-Based Approach for Mining Skyline Periodic Itemset Patterns"

_electronics, doi:10.3390/electronics12234874_

Round 1
Reviewer 1 Report
Comments and Suggestions for Authors
SUMMARY
This paper introduces Bitwise Skyline Periodic Itemset Pattern Mining (BitSPIM) as an efficient approach for mining Skyline Periodic Itemset Patterns (SPIPs). It streamlines the SPIP identification process by using bitwise representations and an Apriori-like algorithm, BitTableFI. A novel pruning mechanism is incorporated to enhance efficiency. The experimental results demonstrate that BitSPIM outperforms the existing SPIM method in terms of both time efficiency and memory usage.
COMMENTS
The paper presents an intriguing proposal with the potential for significant advancement over the current state-of-the-art approach in mining Skyline Periodic Itemset Patterns (SPIPs). The experimental results indeed indicate the superiority of Bitwise Skyline Periodic Itemset Pattern Mining (BitSPIM). However, there are some notable issues in the paper's structure and presentation that should be addressed to enhance its readability and comprehension:
- The introduction should be more self-contained. It currently assumes a high level of prior knowledge, making it challenging for readers new to the subject to understand the context. Providing additional context and motivation for the research would greatly benefit the paper.
- The concept of transactions is unclear. Providing clear definitions and examples to elucidate this concept will help readers better grasp the paper's core ideas.
- The background section could benefit from more depth and detail. It should include illustrative examples and explanations to make the concepts more accessible to a broad audience. This will not only aid in understanding but also make the paper more informative.
- The paper would benefit from the inclusion of a dedicated section discussing related work, specifically focusing on approaches that have employed bit representation, and bitwise operations, for the mining of metadata. For instance 10.1109/ICDE55515.2023.00111, 10.1016/j.procs.2015.06.048, and 10.1007/s10489-2018-1182-6 should be reviewed and discussed in this section.
- More details could be provided on the data structures and algorithm steps. The pseudocode provides a high-level overview, but more explanation of implementations may help comprehension.
- The paper mainly focuses on the technical aspects of the proposed algorithm and its performance evaluation through simulated experiments. However, it lacks real-world application examples or case studies that demonstrate the practical implications and potential benefits of mining SPIPs using BitSPIM in real-world scenarios.
- It's important to ensure that the positioning of images and tables in the paper aligns with the sections where they are discussed for better clarity and comprehension. For instance, moving Figure 3 before Table 6 is a logical adjustment that can improve the flow of the paper and enhance the reader's understanding of the content.
Comments on the Quality of English LanguageThe English language's quality meets acceptable standards.
Reviewer 2 Report
Comments and Suggestions for Authors
Dear Authors,
Thanks for submitting your paper. Please readd the following comments carefully:
1- the citation ends by the end of introduction section, however, there are more equations, algorithms, and some parts of the work. It should be clarified more which parts of the work was developed by you or it is already created by you.
2- One of the major points which might make the work much nicely presented is to demonstrate and end to end example of your proposed approach. It would be much better to do the same to the state-of-the-art or the baseline approach that you evaluated your algorithm against.
3- it would be better to expand the discussion to include the impact of your approach as table against the state-of-the-art with some examples of the structure for both models.
4- You could provide a small examples for each section and how it is done. i.e., section 3.3.1, you could produce an example of how the identification process was done after the modification.
Over all, the proposed methodology was nicely presented and I hope the proposed notes could help to make some improvements ...
best,
The reviewer...
Reviewer 3 Report
Comments and Suggestions for Authors
In this paper, the authors conducted a comprehensive comparison between BitSPIM and SPIM, focusing on average memory usage (MB) and running time (ms). The initial presentation in the abstract emphasized the superior performance of BitSPIM over SPIM, particularly highlighting its significant outperformance in memory usage & time.
Upon closer examination of the results across the 10 datasets, namely T25I10D10K, C20D10K, and BMS-Webview-1, it becomes apparent that BitSPIM's running time is competitive and close to SPIM in these specific instances. The datasets T25I10D10K, C20D10K, and BMS-Webview-1 exhibited comparable running times between BitSPIM and SPIM (in Figure 3).
To provide a broader perspective on the performance, the authors should explicitly state in which datasets BitSPIM achieved higher running times, identify the dataset that is closest in performance, and acknowledge instances where BitSPIM did not perform as well.
Furthermore, to bolster the credibility and significance of their findings, the authors should extend their comparison to include results from other similar approaches. A thorough examination of how BitSPIM performs in relation to existing methods on these datasets would enhance the persuasiveness of their proposed method. This comparative analysis with other researchers' works will add weight to the proposed approach, providing a more robust foundation for their claims.
Round 2
Reviewer 1 Report
Comments and Suggestions for Authors
I would like to thank the authors for following the suggestion I highlighted in my previous review. After reviewing the revised version of the manuscript, I found that the current version of the manuscript has significantly improved in clarity and overall comprehensibility for a broader audience. To this end, I believe that the paper could be accepted in its current form.
Reviewer 3 Report
Comments and Suggestions for Authors
Author addressed the some previous concerns.